# Role of Chest Imaging in Viral Lung Diseases

**DOI:** 10.3390/ijerph18126434

**Published:** 2021-06-14

**Authors:** Diletta Cozzi, Eleonora Bicci, Alessandra Bindi, Edoardo Cavigli, Ginevra Danti, Michele Galluzzo, Vincenza Granata, Silvia Pradella, Margherita Trinci, Vittorio Miele

**Affiliations:** 1Department of Emergency Radiology, Azienda Ospedaliero-Universitaria Careggi, 50134 Florence, Italy; eleonora.bicci92@gmail.com (E.B.); bindi.alessandra@gmail.com (A.B.); edoardocavigli@yahoo.it (E.C.); ginevra.danti@gmail.com (G.D.); pradella3@yahoo.it (S.P.); vmiele@sirm.org (V.M.); 2SIRM Foundation, 20122 Milan, Italy; v.granata@istitutotumori.na.it; 3Department of Emergency Radiology, San Camillo Forlanini Hospital, 00152 Rome, Italy; galluzzom@tiscali.it (M.G.); margherita.trinci@libero.it (M.T.); 4Istituto Nazionale Tumori IRCCS “Fondazione G. Pascale”, 80100 Naples, Italy

**Keywords:** coronavirus, COVID-19, viral pneumonia, computed tomography, differential diagnosis

## Abstract

The infection caused by novel beta-coronavirus (SARS-CoV-2) was officially declared a pandemic by the World Health Organization in March 2020. However, in the last 20 years, this has not been the only viral infection to cause respiratory tract infections leading to hundreds of thousands of deaths worldwide, referring in particular to severe acute respiratory syndrome (SARS), influenza H1N1 and Middle East respiratory syndrome (MERS). Although in this pandemic period SARS-CoV-2 infection should be the first diagnosis to exclude, many other viruses can cause pulmonary manifestations and have to be recognized. Through the description of the main radiological patterns, radiologists can suggest the diagnosis of viral pneumonia, also combining information from clinical and laboratory data.

## 1. Introduction

Viral pneumonia is one the most frequent respiratory diseases among very young people and the elderly, contributing to an increase in the number of hospitalizations and deaths, mainly in subjects over 60 years of age [1,2]. Many risk factors can predispose to the onset of a viral respiratory tract infection, such as immune system disorders, malnutrition in children, and tobacco smoking and chronic obstructive pulmonary disease (COPD) in adults [3]. One of the major risk factors is immune system impairment; in fact, immunocompromised patients have a higher risk of developing a viral infection because of the decreased resistance and response to it [4]. The immunocompromised patient category is broad, considering the high number of people with immunity deficits, mainly due to the advent of new therapies (including long-term treatments with immunosuppressive drugs, chemotherapies, and steroids) [5,6]. The symptoms of viral infections are quite similar and usually non-specific despite the different aetiologies, including cough, fever, and dyspnoea, making a differential diagnosis difficult and dependent on the immune status of the host [7]. The pathogens responsible for these diseases can be classified according to the immunocompetent or compromised status of the patient, more predisposed to pneumonia induced by specific viral agents, being divisible in atypical pneumonia in otherwise healthy hosts, and viral pneumonia in immunocompromised ones [8]. The most common respiratory viruses in both categories are influenza, human parainfluenza (HPIV), adenovirus and respiratory syncytial virus (RSV), but in immunocompromised people there can also be infections due to cytomegalovirus (CMV), herpes simplex virus (HSV), and Epstein–Barr and Varicella-zoster viruses [9,10]. Despite recent advances in diagnostic methods such as culture, rapid antigen testing, polymerase chain reaction (PCR) testing, and serologic analysis, specific viral diagnosis often remains difficult [11]. Although radiologic imaging alone is not sufficient for the diagnosis, imaging is essential in association with clinical and laboratory testing. To assess the presence of pulmonary involvement and extension of disease, diagnostic imaging techniques are traditional chest radiographs (CXR) and computed tomography (CT). CXR may show a negative radiological pattern or otherwise areas of monolateral or bilateral consolidations, in association with nodular opacities, bronchial wall thickening, and small pleural effusions. Lobar consolidation is uncommon in patients with viral pneumonia, being more characteristic of bacterial forms [12]. Chest CT is the gold standard for the evaluation of viral pneumonia showing different patterns, sometimes with nonspecific imaging findings [12]. This review focuses on the imaging patterns of viral pneumonia caused by different pathogens, including new viruses that have caused severe pandemics in recent years, most recently COVID-19. Furthermore, this review may offer diagnostic imaging insights that are useful for differential diagnosis in the radiologist’s daily work; moreover, the present work is based on original images acquired by all the authors and the microbiological diagnosis is confirmed mainly on broncho-alveolar lavage (BAL). Table 1 summarizes the main CXR and CT patterns of the viral pneumonia from specific viruses described in the article’s images. (Table 1).

## 2. Lower Respiratory Tract Viral Infections in Immunocompetent

### 2.1. Coronaviridae

#### 2.1.1. SARS-CoV-2—Coronavirus Disease 2019 (COVID-19) 

SARS-CoV-2 (severe acute respiratory syndrome coronavirus 2) began to expand in 2019 in China and spread worldwide, with the first reported cases in Italy in February 2020. This virus belongs to the coronavirus family and is highly infectious, being able to cause severe pneumonia up to acute respiratory distress syndrome (ARDS). Its recognition is very important for a prompt diagnosis, involving real-time reverse transcription-polymerase chain reaction (RT-PCR) of viral nucleic acid [13]. Despite this being the gold standard, there are some limitations due to the possibility of a false-negative test and delayed results. Therefore, radiological imaging is mandatory to have a rapid evaluation of thoracic involvement. Although CT is much more sensitive in detecting radiological patterns in coronavirus disease-19 (COVID-19) pneumonia, most Italian hospitals are using CXR as the first line of investigation as the Italian Society of Radiology (SIRM) recommends using CXR as a first-line imaging tool, leaving chest CT to other roles such as the identification of COVID-19 pneumonia typical features in particular cases with specific issues such as clinical-radiological discordance, acute complication (i.g., pulmonary embolism or severe respiratory failure), or after intubation before transporting the patient to an intensive care unit [14,15,16,17]. As Cozzi et al. reported in their work involving 482 patients with clinical-epidemiological suspect for COVID-19 and RT-PCR and CXR performed at admission to the emergency department, the most highlighted pulmonary patterns were lung consolidations, ground-glass opacities (GGO), nodules and reticular–nodular opacities, manifesting as interstitial pneumonia with diffuse alveolar damage [18] (Figure 1). Moroni et al. also attempted to evaluate the diagnostic accuracy of CXR in 327 patients during the descending phase of the pandemic, finding that it showed a specificity of 83% and sensitivity of 60% and that the findings more often highlighted were GGO and diffuse or bibasal distribution. In addition, CXR showed a good correlation with outcome in these patients [19]. Typical CT findings of COVID-19 pneumonia in the early phase are predominantly peripheral, bilateral GGOs, in association with limited consolidations, interlobular and intralobular septal thickening creating a “crazy-paving” pattern (Figure 2). Air bronchograms, vascular enlargement, halo sign, and reverse halo sign are also reported [20,21,22,23,24]. As the disease proceeds, the GGOs spread to more diffusely involve the lung lobes with a marked increase in the associated consolidation component. In a study conducted by Li et al., only 2 of 51 patients did not show GGO or consolidation at CT exam. Therefore, GGO and consolidation were, as previously stated, two main signs of COVID-19 lesions [25]. Other CT findings that were highlighted in various studies are pleural changes, referring especially to pleural thickening. This was found in 32% of patients, unlike pleural effusion, which was evaluated in 5% of patients and is more typical of other viral coronavirus infections such as MERS [26]. Lymphadenopathy was seen in 4–8% of patients with COVID-19 and was a risk factor for severe or critical disease [27,28]. Pericardial effusion was seen in 5% of patients, likely due to concomitant inflammation. This manifestation was also associated with severe and critical degrees of disease [28]. Of special reference to ultrasound imaging, that in this pathology has shown some typical sonographic features, as in the stage of early disease and before the manifestation of symptoms, is the development of localized vertical irregular artefacts, called B-lines [29]. Peng et al. examined 20 patients with COVID-19 with lung ultrasonography describing multi-lobar distribution patterns: focal B-lines were the main initial feature, followed by alveolar interstitial syndrome in progressive stages, then followed by A-lines during convalescence [30]. This infection shows characteristic ultrasonographic manifestations with bilateral involvement and predominant distribution in the posterior and inferior part of the lungs: COVID-19 generally manifests with initial involvement of the terminal alveoli, adjacent to the pleura, which can be observed by lung ultrasonography [31]. Although artificial intelligence (AI) cannot currently be used as the first detection modality, especially in emergencies, due to medical-legal, ethical, and outcome variability issues, there is much expectation regarding the role it could take in the diagnosis and treatment of serious and complex diseases such as coronavirus [32,33,34]. Grassi et al. used AI software for quantification of pneumonia lesions to facilitate CT diagnosis with the hope that in the future this may be useful to identify the characteristic patterns of these diseases in order to select the best therapy and management for patients [35].

**Figure 1 ijerph-18-06434-f001:**
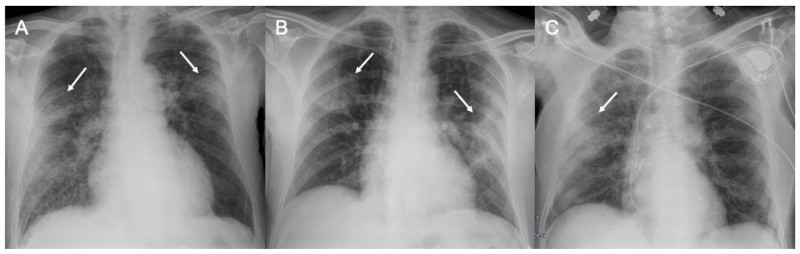
Chest radiograph in COVID-19 pneumonia. Three cases of supine chest X-ray with subpleural consolidations (arrows), in (**A**,**B**) with bilateral involvement, and in (**C**) with main involvement of the right lung.

**Figure 2 ijerph-18-06434-f002:**
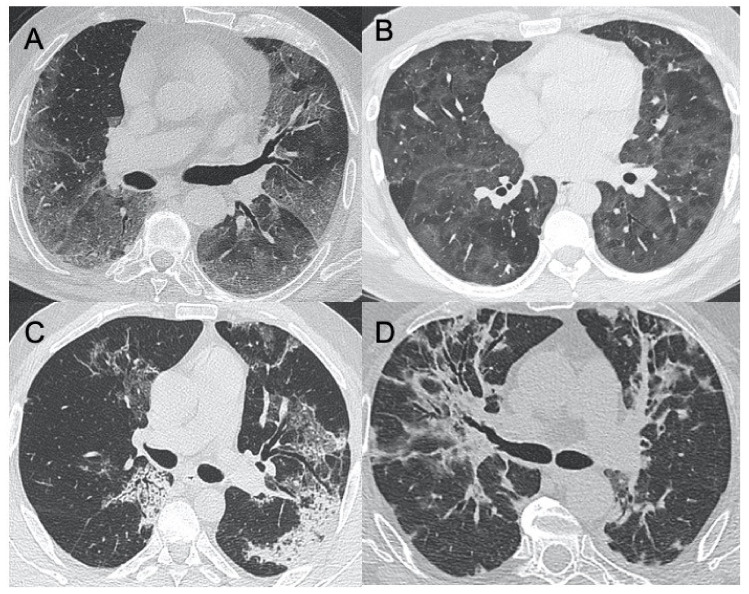
HRCT (High resolution computed tomography) in COVID-19 pneumonia. Diffuse ground-glass opacities involving both lungs (**A**) and with peri-lobular pattern (**B**) in the acute phase of the infection. (**C**,**D**) show two cases of sub-acute interstitial pneumonia, with decrease in ground-glass opacities and the presence of subpleural focal consolidations and thickening of the interlobular/intralobular interstitium.

#### 2.1.2. Middle East Respiratory Syndrome (MERS)

MERS is an illness caused by the Middle East respiratory syndrome coronavirus (MERS-CoV) belonging also to the coronavirus family. It was first found in Saudi Arabia in September 2012, reaching the maximum number of countries in 2015 [36]. Clinically, it manifested itself from asymptomatic forms to disease primarily involving the upper respiratory tract, leading to pneumonia, respiratory failure, and multiple organ failure (MOF) resulting in death, reaching a mortality rate that approaches 60% [37]. On CXR, the main chest radiologic finding associated with MERS-CoV is ground-glass opacity followed by consolidation that can be patchy, confluent, or can appear as nodular areas of opacity. Unifocal involvement (69%) is more common than multifocal involvement (30%) [38]. MERS pneumonia appears on CT images as subpleural and lower lesions, with extensive GGO and consolidation. One particular feature that has been found in patients with this viral pneumonia is pleural effusion, which has been seen to correlate with negative prognostic factors [12,39]. 

#### 2.1.3. Severe Acute Respiratory Syndrome (SARS)

SARS also belongs to the coronavirus family and was first discovered in China in 2002 [40]. After incubation of approximately 2–10 days, it manifested clinically with flu-like symptoms and dyspnoea. Radiological features were similar to those of other viral pneumonias, with areas of consolidation involving mainly the lower portions of the lung. Monofocal manifestations were more common than multifocal or bilateral ones as in MERS. In contrast, pleural effusion was rarely seen [41,42]. This is probably because they belong to the same family of Coronaviridae and therefore have some pathological mechanisms in common; SARS, MERS, and COVID-19 certainly have similar radiological features. However, some differences were also highlighted, such as the involvement of both lungs on imaging of the early stages of the disease is more frequent in patients with COVID-19 than in SARS and MERS. Also, pleural effusion, cavitations and lymphadenopathies are more frequent in MERS, where pleural effusion is often present and is a negative prognostic index in SARS [12,38]. As previously stated, lymphadenopathy was seen in 4–8% of patients with COVID-19 and was associated with higher risk for severe disease [27,28]. Regarding SARS and COVID-19, some differences were highlighted in the tendency of distribution of GGO areas, being more often multifocal and peripheral in the upper lung lobes and basilar in the lower lobes in COVID-19. Moreover, as previously mentioned, both SARS and MERS tend to be mainly unilateral and features such as halo or reversed halo signs have not been reported in literature; nevertheless, Li et al., in their studies on COVID-19 infections, reported a 17.6% rate of halo sign and a 3.8% rate of reversed halo sign [43,44]. 

#### 2.1.4. Orthomyxoviridae

##### Influenza A, B, and C

The influenza virus belongs to the Orthomyxoviridae family. It generally manifests itself during the winter period, causing infections of the upper airways and, in patients with chronic diseases and in the elderly or immunocompromised, may lead to the onset of pneumonia [10]. The characteristics at CXR are bilateral reticulonodular opacities in association with areas of consolidation, usually in the lower lobes. Lobar consolidation may suggest bacterial superimposed infection [45,46]. However, it has been shown that these features are different in immunocompromised patients affected by influenza. In fact, according to two studies conducted by Leung and Oikonomou, which showed that in the enrolled patients, at CT imaging, the manifestations of the virus were mainly represented by GGOs with consolidation, bilateral patchy consolidation, and ill-defined small nodules [47,48] (Figure 3). In contrast, the same differences between influenza pneumonia in immunocompetent and immunocompromised patients were not found in the study conducted by Kloth et al., who retrospectively evaluated CT patterns of lung infiltration caused by different influenza virus strains in the two groups of patients and did not assess significant differences. This underlines once again how these pulmonary patterns of viral infection are very often difficult to evaluate [49]. Two special references should be made regarding the H1N1 and H5N1 forms of influenza. These have led to epidemics with high mortality rates. However, unlike the common flu, they presented as rapidly progressive pneumonia that very often led to acute respiratory distress syndrome. CT imaging features were multifocal consolidations and diffuse areas of GGO. Lymphadenopathy, cavitations, pleural effusion, and pneumatocele were also observed [50,51]. Valente et al., in their study conducted on 50 patients who had developed a severe form of influenza A (H1N1), reviewed CXR and CT imaging, evaluating how the most frequently found features were unilateral or bilateral GGO with or without associated areas of consolidation with predominantly peribroncho-vascular and subpleural distribution [52] (Figure 4). It is important to underline the complications of this pathology because post-ARDS pulmonary fibrosis is a possible consequence. In their study, Mineo et al. evaluated how, out of 20 patients who had contracted the H1N1 virus with consequent development of pneumonia and severe clinical and radiological conditions, 25% had experienced ARDS and 10% of these developed fibrosis [53]. Differentiating between COVID-19 pneumonia and influenza is very complicated given the large overlap in radiological features. Lin et al., in their study on CT imaging of 52 patients with COVID-19 and 45 with influenza virus, found that parenchymal lesions in the former patients were predominantly peripheral and close to the pleura, whereas lesions from influenza virus pneumonia were randomly distributed. Moreover, in the latter group, it was more likely to find mucoid impactions and pleural effusion [54].

**Figure 3 ijerph-18-06434-f003:**
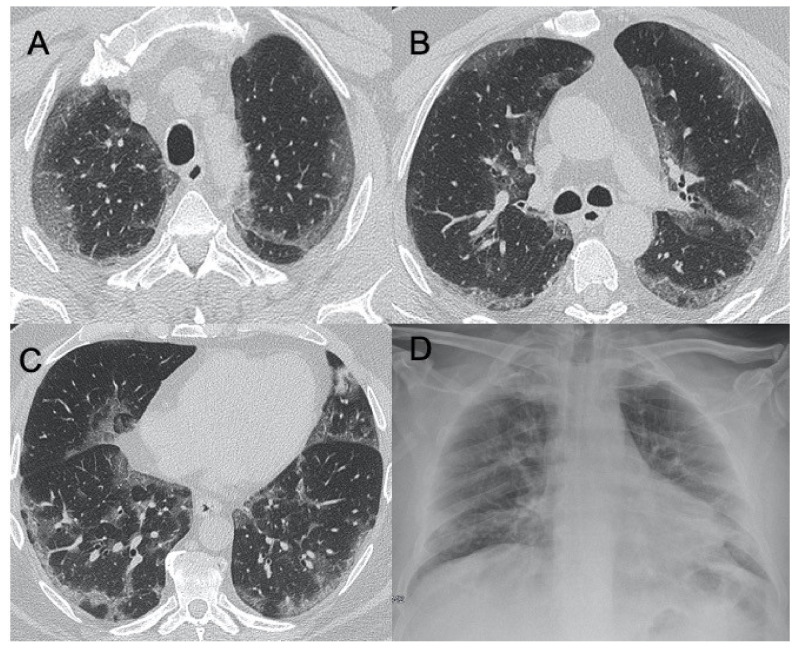
Influenza A. Figures in (**A**–**C**) show a case of influenza A with an interstitial pattern very similar to COVID-19: ground-glass opacities are mainly subpleural and bilateral, with a peri-lobular pattern of distribution. Figure (**D**) is a supine chest radiograph of the same patient, with diffuse interstitial involvement.

**Figure 4 ijerph-18-06434-f004:**
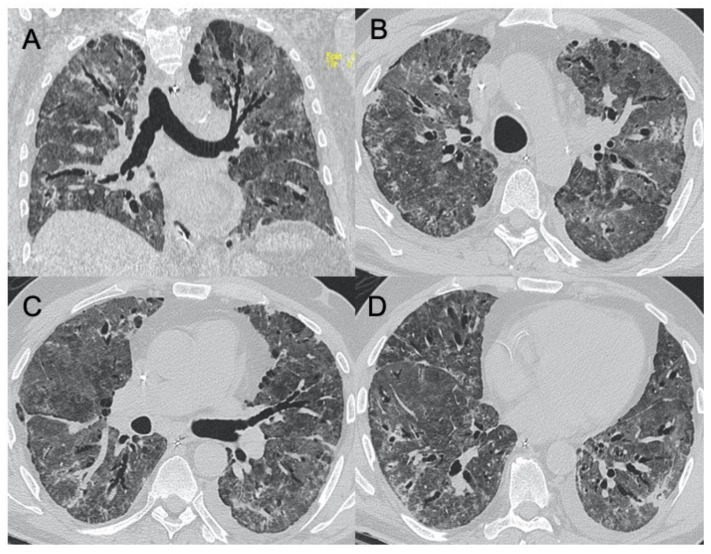
H1N1 interstitial pneumonia. These images (**A**–**D**) show a case of H1N1 related-pneumonia complicated in acute respiratory distress syndrome (ARDS), with diffuse and bilateral ground-glass opacities and traction bronchiectasis/bronchiolectasis.

#### 2.1.5. Adenoviridae

Adenovirus is a DNA virus, responsible for 5–10% of respiratory infections in children [55]. This manifests radiologically as bilateral and multifocal GGO and may show lobar or segmental involvement, similarly to bacterial pneumonia. Long-term complications of pneumonia include bronchiectasis, bronchiolitis obliterans, and unilateral hyperlucent lung, with Swyer–James–Macleod syndrome, which seems to be the result of acute infection in infancy or early childhood that damages the terminal and respiratory bronchioles and prevents their normal development [56,57]. On CXR, it is characterized by a unilateral small lung with hyperlucency and air trapping on expiration. At CT, the affected lung may undergo structural subversion in its entirety, but there may also be lobar, segmental, or sub-segmental involvement (Figure 5). The lung presents as hyperlucent with reduced vascularity and the possibility of bronchiectasis [58].

**Figure 5 ijerph-18-06434-f005:**
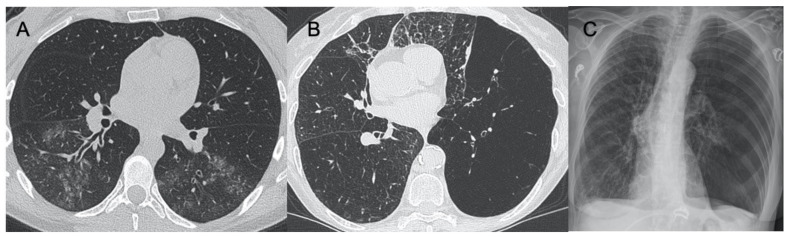
Adenovirus pneumonia and Swyer–James syndrome. Figure in (**A**) shows a case of acute adenovirus pneumonia, with typical multifocal and lobar ground-glass opacities, similar to bacterial pneumonia. Images in (**B**,**C**) show a case of long-term complication, a unilateral hyperlucent lung (Swyer–James–MacLeod syndrome).

#### 2.1.6. Pneumoviridae

##### Respiratory Syncytial Virus (RSV)

RSV is a virus belonging to the Pneumoviridae family and is the most frequent cause of hospitalization in children and also causes disease in the elderly [59]. The virus usually leads to bronchiolitis pneumonia and asthma in all age groups. At the CT investigation, it presents with an airway-centric distribution, with areas of tree-in-bud opacity and bronchial wall thickening, with or without broncho-vascular consolidation [60]. 

## 3. Lower Respiratory Tract Viral Infections in Immunocompromised

### 3.1. Herpesviridae

The herpes virus family is composed of DNA viruses, which can cause both acute infections and remain latent in tissues, causing chronic infections. Although HSV infections can be found in both immunocompetent and non-immunocompetent patients, associated pneumonia occurs predominantly in immunocompromised subjects.

### 3.2. HSV Pneumonia 

HSV pneumonia is predominantly caused by type 1 virus and, more rarely, type 2 [61,62]. HSV is often isolated from the oropharynx and upper respiratory tract of both immunocompetent and immunocompromised people, being in the latter often found as a lower area pathogen [63]. The main risk factor is immunosuppression, such as in the case of cancer, organ transplantation, immunosuppressive therapy, renal failure, and acquired immunodeficiency syndrome (AIDS) [63,64]. Infection manifests clinically with fever, productive cough, dyspnoea, and upper airway obstruction symptoms caused by pseudo-membranes for tracheal ulcers. HSV infection can manifest with three different forms of pulmonary involvement with necrotizing tracheobronchitis, necrotizing pneumonia, or interstitial pneumonia. In particular, referring to the latter, on CXR investigations it manifests as bilateral areas of consolidation with GGO with lobular, segmental, or sub-segmental distribution [65,66]. On CT imaging, multifocal areas of segmental or sub-segmental GGO are observed (Figure 6). Pleural effusion is frequent [61]. In the study by Chong et al., who retrospectively evaluated the CTs of five patients with HSV pneumonia, no differences in radiological manifestations were found between immunocompetent and immunocompromised patients [67].

**Figure 6 ijerph-18-06434-f006:**
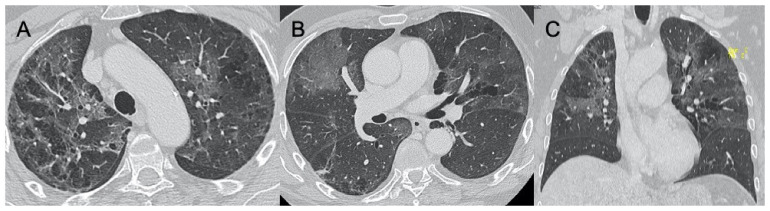
Herpes virus pneumonia. A case of HSV (Herpes virus) (**A**–**C**) pneumonia with bilateral ground-glass opacities with patchy distribution, mainly in both upper lobes.

### 3.3. Varicella-Zoster Virus Infection

Varicella-zoster virus belongs to the herpes virus family. It normally manifests itself as a self-limiting benign infection that predominantly affects the pediatric age, although it may result in a mortality rate of 9–50%, with pneumonia as the most frequent and severe complication. As in other respiratory tract infections, the possibility of severe disease increases with immunocompromised states, haematological malignancies, and pregnancy [68]. Clinically, it manifests with fever, typical rush, cough, and dyspnoea. Although the viral infections of the herpes family show similar aspects on radiological imaging, an exception is made for chickenpox. CXR findings of varicella-zoster virus pneumonia are multiple nodules (5–10 mm) with defined margins that may tend to confluence. Pleural effusion, as well as lymphadenopathy, may be present although they are not common. As the skin manifestations improve, parenchymal nodules tend to disappear in the following weeks. On CT, characteristics are well-defined nodules (1–10 mm) with a halo of GGO. These lesions may calcify and millimetric calcifications may remain in the lung parenchyma (Figure 7). However, unlike the calcifications that can be seen in cases of tuberculous infection or pneumoconiosis, those due to varicella tend to be smaller (2–3 mm), numerous, with defined margins, and randomly distributed in the parenchyma [69,70].

**Figure 7 ijerph-18-06434-f007:**
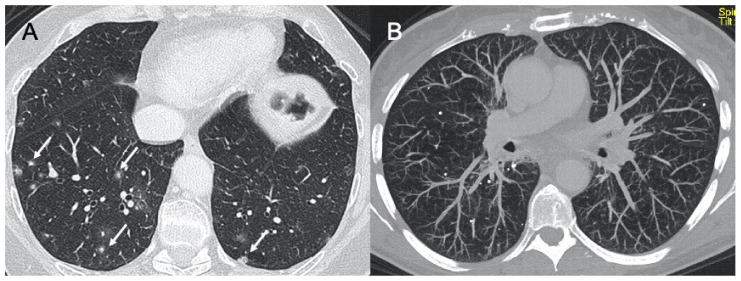
Varicella. A case of acute varicella pneumonia (**A**) with focal nodular consolidations (arrows) and its chronic form with small, tiny calcifications, well-visible in Maximum Intensity Projection (MIP) reconstruction (**B**).

### 3.4. Cytomegalovirus (CMV) Pneumonia

CMV belongs to the herpes virus family, generally responsible for infections that are asymptomatic or with moderate flu-like manifestations in immunocompetent patients. However, this virus can cause pulmonary infections in immunocompromised patients, especially after transplantation, immunosuppressive therapies, and in diseases such as AIDS. The most frequent radiological findings on CT examination are interstitial and alveolar infiltrates as bilateral and asymmetric GGO areas in association with areas of parenchymal consolidation [70] (Figure 8). McGuinness et al., in their work evaluating the CT findings of 21 patients affected by AIDS, assessed that the imaging was mainly represented by ground-glass areas with consolidations and bronchial wall thickening [71]. Kang et al. and Moon et al. both conducted studies on the evaluation of CT findings in immunocompromised non-AIDS patients. Their studies were contrasting, therefore revealing heterogeneity in the possible manifestations of pathology. In particular, in Kang’s study the most frequent patterns were small nodules and areas of parenchymal consolidation, while in Moon’s study bilateral patchy areas of ground-glass opacity were seen in all included patients [72,73]. Differential diagnosis between CMV pneumonia or Pneumocystis jirovecii pneumonia (PJP) is difficult, especially in the early stages of the disease, as the patterns are largely overlapping. Since these pulmonary infections are very common in immunocompromised patients, and the most frequent among infectious complications in patients with AIDS, their differentiation and diagnosis is very important [74]. In the study by Vogel et al., small centrilobular nodules of GGO and consolidations are more indicative of CMV, whereas an apical distribution and the presence of mosaic patterns are more suggestive of PCP [75].

**Figure 8 ijerph-18-06434-f008:**
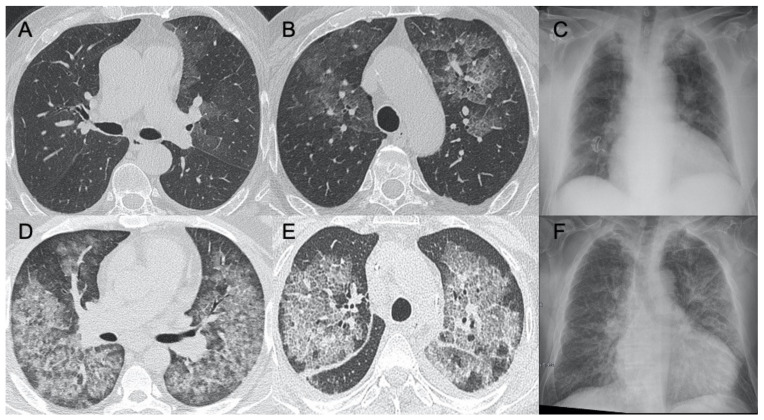
Cytomegalovirus pneumonia. Images in (**A**–**C**) show a case of mild parenchymal involvement on CMV pneumonia. Figures (**D**,**E**) show a patient with diffuse ground-glass opacities with a crazy-paving appearance, visible also in the chest radiograph (**F**).

### 3.5. Epstein–Barr Virus (EBV) Pneumonia

EBV typically infects B lymphocytes and pharyngeal epithelial cells, causing mononucleosis infection: it usually manifests with fever, weakness, tonsillar pharyngitis, and lymphadenopathy often accompanied by splenomegaly [76]. Pulmonary involvement is rare [77]. The most common radiologic CT manifestations are lymphadenopathies and less frequently interstitial infiltrates with diffuse GGOs and consolidations [12] (Figure 9).

**Figure 9 ijerph-18-06434-f009:**
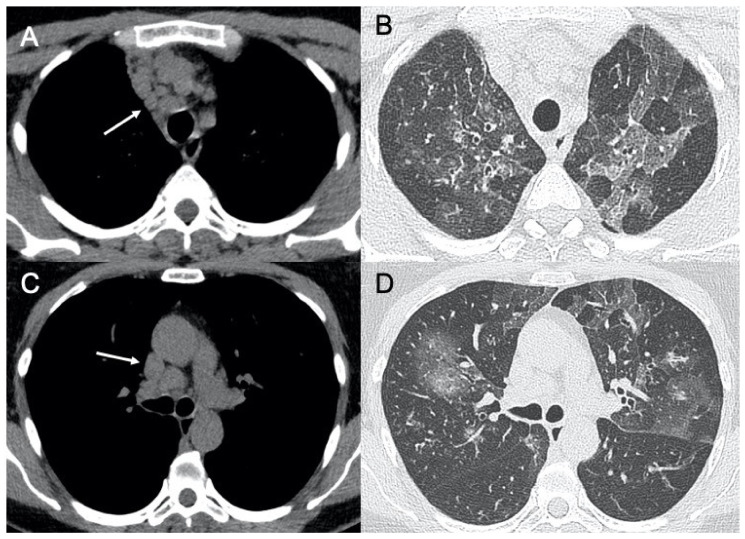
Epstein–Barr virus (EBV) pneumonia. A case of EBV pneumonia showing mediastinal lymphadenopathies (arrows in (**A**,**C**)) associated with focal, lobular ground-glass opacities in both lungs (**B**,**D**).

### 3.6. Paramyxoviridae

#### 3.6.1. Human Parainfluenza Virus (HPIV)

HPIV is an RNA virus belonging to the Paramyxoviridae family and is one of the most common causes of upper respiratory tract infection in adults and children with a typical seasonal pattern. There are four serotypes, although the last three are the major cause of disease [78]. In addition to respiratory manifestations, these viruses cause otitis media, conjunctivitis, and pharyngitis. Particularly severe are the infections caused by HPIV in immunocompromised patients with hematological malignancies or stem cell transplantation, being often the cause of hospitalization and admission to intensive care [79,80]. CT imaging is represented by multiple areas of GGO consolidation in association with centrilobular nodules and bronchial wall thickening [81].

#### 3.6.2. Measles

Measles is an RNA virus belonging to the Paramyxoviridae family. It was one of the major causes of infection among pediatric patients, causing 2–3 million deaths per year. Thanks to the advent of the vaccine, these numbers have been significantly reduced, but it remains a cause of infection capable of causing serious complications, especially in immunocompromised patients and pregnant women [82,83]. Even in Italy, we have seen new measles outbreaks over the years [84]. Symptoms are characterized by fever, typical maculopapular skin rash, cold, cough, and conjunctivitis. Small white papules in the buccal mucosa, the Koplik’s spots, are pathognomonic and usually appear from 1 to 2 days before the disease manifests [85]. Severe forms of measles can cause pneumonia, gastroenteritis, and encephalitis. Pneumonia, in particular, is among the most common complications, with an estimated incidence between 3% and 57% [86]. CXR is the diagnostic imaging for an initial determination of pulmonary involvement, with the possible finding of consolidation or reticular opacities [8,12]. CT findings are nonspecific and include areas of GGO, centrilobular nodules, bronchial and interlobular septal thickening. Pleural effusion and lymphadenopathy were also noted [70]. In the study by Albarello et al. conducted during the 2016–2017 measles outbreak in Italy, CXR imaging of 290 patients was evaluated, respectively. In particular, the most frequently assessed pattern at CXR was bronchial wall thickening, observed in 88.5% of patients. In cases where the suspicion of pneumonia was not supported by CXR imaging, CT investigations and quantitative analysis were also performed to study lungs and airways by quantification of Hounsfield units (HU) and a color-coded display of the thresholds within the lungs. On CT investigations, areas of parenchymal consolidation and areas of GGOs and bronchial/bronchiolar wall thickening were detected. With further quantitative analysis, the actual extent of areas of GGOs and bronchial involvement was assessed [87].

## 4. Conclusions

Last year has been a period characterized by the emergence of a new infection belonging to the coronavirus family, SARS-CoV-2, which has caused a severe worldwide pandemic with the need to develop effective methods in its diagnosis and evaluation, to ensure rapid management and treatment of patients affected by this pathology. Radiology has an important role in this task, being, together with RT-PCR nasopharyngeal and throat swab, the diagnostic method, especially in emergency departments, to provide a rapid assessment of pulmonary involvement. CXR is used in many countries, including Italy, as first-line imaging [15,16]. Many recent studies indicate that CXR may not have the same power of CT, in terms of sensitivity, in the evaluation of pulmonary infection, but it still has an important role in the pandemic, especially for the greater complexity in being able to perform a CT investigation, regarding, for example, the need for disinfection after each examination performed to minimize the possibility of cross-infection, or for the higher radiation doses, especially in young patients [14]. On the other hand, CT has a high sensitivity, around 97–98%, in assessing the presence of abnormalities that may represent pulmonary involvement, although it is not specific to this virus in identifying typical features [88]. CT plays an important role in specific situations such as in the case of acute complications, for example, in the suspicion of pulmonary embolism or severe respiratory failure. Many other viruses belonging to different families in addition to COVID-19 can be responsible for the onset of pneumonia, both in immunocompetent and immunocompromised patients. Because of the great overlapping of manifestations that can occur on imaging, there is little sensitivity in differentiating radiological patterns resulting from viral infections. Although the diagnosis cannot be reached using diagnostic imaging alone, recognition of different patterns of lung involvement may help in differentiation among various viral pathogens. Radiologists can suggest the diagnosis of viral pneumonia, combining clinical and radiographic findings to substantially improve the diagnostic accuracy.

## Figures and Tables

**Table 1 ijerph-18-06434-t001:** Summary table with the main CXR and chest CT features of the viruses explained in the following figures in this review. CT: computed tomography; GGO: ground-glass opacities.

Virus	Chest X-ray Signs	Chest CT Signs	Figure in the Text
Sars-CoV-2	Lung subpleural consolidations, ground-glass opacities, nodules and reticular–nodular opacities, manifesting as interstitial pneumonia with diffuse alveolar damage	In the early phase, predominantly peripheral, bilateral GGOs, in association with limited consolidations, interlobular and intralobular septal thickening creating a “crazy-paving” pattern. Air bronchograms, vascular enlargement, halo sign, and reverse halo sign are also reported	Figure 1 and Figure 2
Influenza A	Bilateral reticular-nodular opacities in association with areas of consolidation, usually in the lower lobes	Multifocal consolidations and diffuse areas of GGO. Lymphadenopathy, cavitation, pleural effusion, and pneumatocele were also observed	Figure 3
H1N1 virus	Unilateral/bilateral GGO with or without associated areas of consolidation with predominantly peribroncho-vascular and subpleural distribution	Unilateral or bilateral GGO with or without associated areas of consolidation with predominantly peribroncho-vascular and subpleural distribution	Figure 4
Adenovirus	Bilateral and multifocal GGO with lobar or segmental involvement, similarly to bacterial pneumonia. In case of complication, unilateral small lung with hyperlucency and air trapping on expiration (Swyer–James Syndrome)	Bilateral and multifocal GGO with lobar or segmental involvement, similarly to bacterial pneumonia	Figure 5
Herpes virus	Bilateral areas of consolidation with GGO with lobular, segmental, or sub-segmental distribution	Multifocal areas of segmental or sub-segmental GGO are observed; pleural effusion is frequent	Figure 6
Varicella Zoster	Multiple nodules (5–10 mm) with defined margins that may tend to confluence. Pleural effusion and lymphadenopathy may be present although they are not common	Well-defined nodules (1–10 mm) with a halo of GGO. These millimetric lesions may calcify.	Figure 7
Cytomegalovirus	Diffuse and bilateral consolidations with interstitial involvement.	Interstitial and alveolar infiltrates, bilateral and asymmetric GGO areas in association with areas of parenchymal consolidation	Figure 8
Epstein-Barr virus	Lymphadenopathies with smooth interstitial parenchymal involvement in both lungs	Lymphadenopathies and less frequently interstitial infiltrates with diffuse GGOs and consolidations	Figure 9

## Data Availability

The data presented in this study are available on request from the corresponding author.

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
