# Peer review of "Role of Chest Imaging in Viral Lung Diseases"

_ijerph, 2021, doi:10.3390/ijerph18126434_

Round 1
Reviewer 1 Report
The authors performed a review of the role of chest imaging in viral Lung diseases and the imaging patterns of viral pneumonia caused by different pathogens, including new viruses that have been causing severe pandemics in recent years, most recently COVID-19. Overall, the paper looks good but it seems that the manuscript has been written hastily. A complete proof-reading is highly required; the formatting must be corrected, the references included, and typos corrected.
-A tabulated summary would be very useful, it can include for example the imaging patterns corresponding to each pathogen.
Here is only a small set of typos:
-Delete "." after the title.
-Use ";" to separate the keywords.
-Heading "1. Manuscript" should be 1. Introduction. Same for other headings and subheadings.
-Use "et al." instead of "et al" consistently.
-"the most high-lighted pulmonary pattern were lung" use either patterns or was.
-"as already said before," delete either already or before.
-Put "." at the end of the conclusion.
-If figures have been taken from the existing literature, please give credit.
Author Response
Thank you for the revision. Here the corrections uploaded.

Reviewer 2 Report
Comments to the Author
Summary of paper:
In this manuscript, the authors reviewed the lung imaging manifestations of various diseases including COVID-19, MERs, etc. The publication of this manuscript may help doctors determine the type of disease based on lung images.
Major comments:
CXR and CRX appear in this manuscript, this manuscript does not explain CRX.
Please make sure that the abbreviation in this manuscript is given the full name when it first appears.
I think there are few representative figures of patients’ chest images in this manuscript, please consider adding more representative figures.
I think the author's narrative on the recognition of COVID-19 by artificial intelligence (AI) in the manuscript is relatively simple, and the details should be added appropriately.
I have not seen references in this manuscript.
Author Response
Thank you for the revision. Here our responses.

Round 2
Reviewer 1 Report
The style of and naming of the sections and subsections need to be clear and correct. For example, is the "1. Manuscript" a correct heading?
A tabulated summary is highly recommended to include. It would help reader to better follow the paper.
A careful proof reading prior to submission is strongly recommended.
Author Response
Thank you for the revision. Here our responses.
The style of and naming of the sections and subsections need to be clear and correct. For example, is the "1. Manuscript" a correct heading?
We have followed Authors' Guidelines of the journal. When we'll have the Proofs, we may correct the layout.
A tabulated summary is highly recommended to include. It would help reader to better follow the paper.
Thank you for the advice, but the Authors Guidelines do not include a Summary paragraph in the writing of the manuscript.
A careful proof reading prior to submission is strongly recommended.
Surely it will be done when the Proofs are sent back to the Authors.
Reviewer 2 Report
The authors have revised the manuscript according to the suggestion of reviewers. It can be accepted now.
Author Response
Thank you for your review.
The Authors